# Ultrasound-Guided Approach to the Distal Tarsal Tunnel: Implications for Healthcare Research on the Medial Plantar Nerve, Lateral Plantar Nerve and Inferior Calcaneal Nerve (Baxter’s Nerve)

**DOI:** 10.3390/healthcare12202071

**Published:** 2024-10-17

**Authors:** Alejandro Fernández-Gibello, Gabriel Camuñas Nieves, Ruth Liceth Jara Pacheco, Mario Fajardo Pérez, Felice Galluccio

**Affiliations:** 1Clínica Vitruvio, 28003 Madrid, Spain; gabrielcamunas@gmail.com (G.C.N.); mfajardoperez@yahoo.es (M.F.P.); felice.galluccio@momarc.org (F.G.); 2Podiatry Department, La Salle Centro Universitario, 28023 Madrid, Spain; 3Morphological Madrid Research Center (MoMaRC), Ultradissection Spain EchoTraining School, 28029 Madrid, Spain; licethjara1@hotmail.com; 4Fisiotech Lab Studio, Rheumatology and Pain Management, 50136 Firenze, Italy; 5Center for Regional Anesthesia and Pain Medicine (CRAPM), Wan Fang Hospital, Taipei Medical University, Taipei 116, Taiwan

**Keywords:** foot health, ultrasonography, tarsal tunnel syndrome, medial plantar nerve, lateral plantar nerve, Baxter’s nerve, nerve decompression

## Abstract

Background/Objectives: The tibial nerve, commonly misnamed the “posterior tibial nerve”, branches into four key nerves: the medial plantar, lateral plantar, inferior calcaneal (Baxter’s nerve), and medial calcaneal branches. These nerves are integral to both the sensory and motor functions of the foot. Approximately 15% of adults with foot issues experience heel pain, frequently stemming from neural origins, such as tarsal tunnel syndrome (TTS). TTS diagnosis remains challenging due to a high false negative rate in neurophysiological studies. This study aims to improve the understanding and diagnosis of distal tarsal tunnel pathology to enable more effective treatments, including platelet-rich plasma, hydrodissections, radiofrequencies, and prolotherapy. Methods: Ultrasound-guided techniques were employed to examine the distal tarsal tunnel using the Heimkes triangle for optimal probe placement. Results: The results indicate that the tunnel consists of two chambers separated by the interfascicular septum, housing the medial, lateral plantar, and inferior calcaneal nerves. Successful interventions depend on precise visualization and patient positioning. This study emphasizes the importance of avoiding the calcaneus periosteum to reduce discomfort. Conclusions: Standardizing nerve involvement classification in TTS is difficult without robust neurophysiological studies. The accurate targeting of nerve branches is essential for effective treatment.

## 1. Introduction

The tibial nerve, wrongly called the “posterior tibial nerve” [1,2], has four nerve branches: The medial plantar nerve, a sensitive nerve that superficially innervates the medial plantar area of the foot from the first to the medial area of the fourth toe approximately and motor innervates mainly the abductor hallucis (ABDH), the flexor hallucis brevis (FHB), the flexor digitorum brevis (FDB), and the first and second lumbrical; on the other hand, we have the lateral plantar nerve which, at the superficial sensory level, innervates the lateral plantar area of the foot together with the fourth and fifth toes, and at the motor level, it innervates the quadratus plantae (QP), the lateral head of the FHB, first interosseous, oblique and transverse head of the adductor hallucis (OHADDH-THADDH), lumbricals, second to fourth interosseous, and the short flexor of the fifth toe and opponens of the fifth toe; we continue with the inferior calcaneal nerve, also known as the first branch of the lateral plantar nerve or Baxter’s nerve. At the sensory level, it innervates only deep structures such as the periosteum of the calcaneus, long plantar ligament, etc., and at the motor level, it innervates the abductor digiti quinti (ABD5°), the proximal zone of the FDB, and the lateral head of the QP; and lastly, we have the medial calcaneal branches, sensitive branches that innervate the medial and plantar surface of the heel and medial part of the Achilles tendon [3].

Some authors believe that heel pain occurs in 15% of adults with foot problems, and according to Oztuna, some of this pain may have a neural origin in patients with both acute and chronic pain, potentially indicating tarsal tunnel syndrome (TTS) [4].

It is an under-diagnosed condition because neurophysiological studies have an estimated false negative rate of 50% [4]. According to the American Association of Neuromuscular and Electrodiagnostic Medicine, TTS can be diagnosed if the patient has neuropathic signs or symptoms with a positive Tinel’s sign [5].

However, there are other methods that can assist in the diagnosis of this condition: ultrasound allows us to assess the echostructure of the nerve, and we can also evaluate the cross-sectional area of the nerve before entering the tunnel and within the tunnel as a diagnostic sign.

Due to this diagnostic difficulty and the complexity of the management of this type of pathology, our aim is to improve the knowledge of this condition by focusing on the pathology of the distal tarsal tunnel, so that different specialists can learn to safely approach the nerve branches and thus be able to carry out treatments such as infiltrations, platelet-rich plasma, hydrodissections, radiofrequencies, prolotherapy, etc.

## 2. Sonoanatomy Considerations

As a reminder, the tarsal tunnel has two zones of compression: the proximal tarsal tunnel, where the tibial nerve is compressed by the ankle flexor retinaculum, and the distal tarsal tunnel, where the various branches of the tibial nerve are compressed by the deep fascia of the abductor hallucis.

Anatomically, the distal tarsal tunnel has two chambers, the upper chamber which contains the medial plantar nerve and the lower chamber which contains the lateral plantar nerve and the inferior calcaneal nerve, separated by the interfascicular septum forming an “italic *T*” [6,7]; Figure 1. This is important clinically because these are separate tunnels and the treatments therefore need to be more selective.

To facilitate an ultrasound exploration of the distal tarsal tunnel, we will use the “Heimkes triangle” [6] and position the probe on the line that joins the most posterocentral point of the calcaneus to the tubercle of the scaphoid. We must remember that the nerves at this level make a bend to enter the porta pedis, so in order to obtain a perpendicular ultrasound incidence and improve ultrasound visibility, it is necessary to tilt the probe plantarly (Figure 2).

Another aspect to take into account is that the inferior calcaneal nerve is sometimes positioned under a vein of the lateral plantar neurovascular bundle and we would need to compress the vein to avoid posterior acoustic enhancement and thus improve the visibility of the branch [7].

## 3. US-Guided Approach

For the ultrasound-guided approach, the patient can be positioned in the supine decubitus position with the leg in the “frog position” or in the prone decubitus position, which, although disadvantageous for some patients, allows us to visualize both the needle insertion and the position of the probe. We place the probe on the A-B line of the Heimkes triangle (Figure 1), remembering to tilt the probe plantarly to improve the ultrasound visibility of the nerves.

During the approach, it is recommended to make the needle visible along its entire length, so a proximo-distal approach should be made along the long axis of the probe. It is preferable to avoid approaching too close to the periosteum of the calcaneus in order to reduce patient pain or use subcutaneous anesthesia if necessary.

Often, the needle tilt makes it difficult to see the tip of the needle. If the ultrasound machine is not equipped with needle vision software, we will tilt the needle when we think we are in the chamber and a “tent” effect of the deep fascia of the ABDH should be visible.

The approach will depend on the characteristics of the patient and the preference of the specialist. We are showing an approach in the long axis of the probe in the proximal–distal direction, although a disto-proximal approach is also possible (Figure 3). The main requirement is that we manage to enter the desired chamber without damaging the noble structures.

## 4. Discussion

There are two classifications that standardize the degree of nerve involvement, such as the Seddon and Sunderland classifications. However, this is something we are unable to establish in the tarsal tunnel, as there is no neurophysiological equivalent to the gold standard used in the carpal tunnel. We are not able to establish or approximate the degree of nerve injury and locate the area of involvement and/or compression, determine whether there is distal nerve degeneration or not, etc. This is essential because the degree of injury determines the degree of nerve regeneration and we should know before treatment whether the prognosis is favourable or not [8].

Tarsal tunnel syndrome is a complicated pathology to diagnose because, as we have already mentioned, it has a large number of false negatives and the treatments have a low rate of effectiveness because if we do not know the number of branches affected, the degree of injury to them, etc., then we cannot give the patient the best treatment. This is why more research is needed into neurophysiological protocols that allow us to obtain a more accurate diagnosis, as some authors have described [9], but we must not forget that if we only study the medial and lateral plantar branches, we are only studying 50% of the branches and we are ignoring the inferior calcaneal nerve and the medial calcaneal branch.

### 4.1. Radiofrequency

Pulsed radiofrequency can be an effective treatment, often used in combination with other treatments for the management of recalcitrant plantar fasciopathies, with the aim of neuromodulating or continuous radiofrequency. It can be an effective treatment in the ablation of the inferior calcaneal nerve for the treatment of neuropathic heel pain when Baxter’s nerve compression is involved, or simply to neuromodulate if the patient has nerve involvement of the tibial nerve or one of its branches. Some studies have used pulsed RF on the medial calcaneal branch to treat plantar fasciopathies, but we must remember that this is a superficial sensory branch and it does not innervate the plantar fascia, so in the absence of mononeuropathy of this branch, it would not be the most appropriate treatment [10].

Other authors perform ablative radiofrequency of the medial and inferior branches of the calcaneus to treat chronic heel pain, achieving improvement in 88% of patients [11], but it is important to know that by ablating the inferior branch of the calcaneus, the motor function of the abductor of the fifth toe, part of the flexor digitorum brevis, and the lateral head of the plantar quadratus is eliminated. In some patients, if we obtain a sensory response without a motor response by neurostimulation prior to RF, we may be able to be selective in ablating their sensory branch, but further investigation of the division point of their sensory and motor branches is necessary to perfect the approach.

It is necessary to understand that in tarsal tunnel syndrome, the involvement of a single nerve branch is rare, and that the most common form is mixed involvement in different degrees of injury, so by removing the inferior calcaneal branch for chronic heel pain, if the patient’s tibial nerve is affected in the proximal tarsal tunnel, this will make the treatment less effective.

### 4.2. Ultrasound-Guided Hydrodissection

Ultrasound-guided nerve hydrodissections are performed to relieve pressure and decompress the affected nerves if they are compressed, but as we have already mentioned, it is necessary to find the affected branches through a neurophysiological study or, failing that, use the points that show a positive Tinel’s sign. If the patient has a positive Tinel’s sign or neuropathy at the level of the distal tarsal tunnel, we will perform an upper and/or lower chamber hydrodissection to decompress the area. Regarding Baxter’s nerve or the inferior calcaneal nerve, we should be aware that in some patients, it may have its own chamber, as described in this anatomical study [12].

Neural hydrodissection techniques could be an effective long-term treatment because they specifically target one of the most common causes of tarsal tunnel syndrome: nerve compression. However, it is essential to understand that the origins can be multifactorial and that the high rate of false negatives in nerve conduction studies complicates the diagnosis and prognosis of this condition. Depending on the degree of nerve injury, regeneration can be complete, partial, or virtually absent. Therefore, it would be beneficial for future research to focus on improving the neurophysiological diagnosis of this syndrome, aiming to establish a ‘gold standard’ test that provides a more accurate diagnosis and a better understanding of the extent of the injury.

## 5. Conclusions

In conclusion, we have to understand the complexity of the tarsal tunnel and be as accurate as possible in our diagnosis in order to provide the most appropriate treatment and give our patients a more realistic expectation of improvement. Further research is needed to improve diagnosis and quantify the degree of injury.

## Figures and Tables

**Figure 1 healthcare-12-02071-f001:**
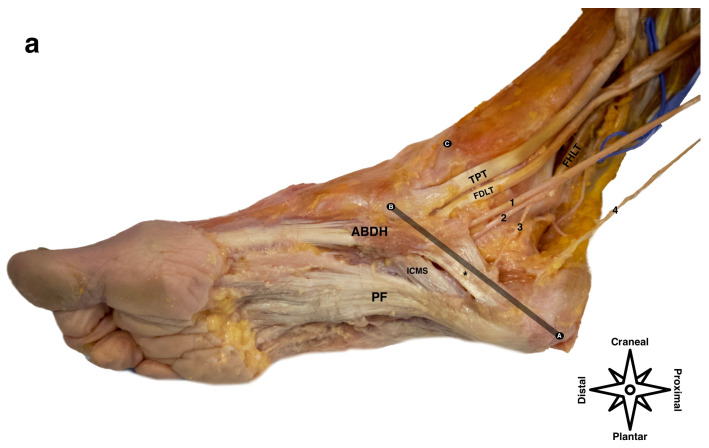
Image (**a**) shows the dissection of a tarsal tunnel in which the laciniate ligament and the proximal compartments have been removed, leaving the distal tunnel with the key structure (deep fascia of the hallux abductor) with an asterisk. Points A, B, and C show the Heimkes triangle and line A-B shows the area where the ultrasound probe is to be positioned. ABDH (abductor hallucis), PF (central component of the plantar fascia), ICMS (intercompartmental medial septum), TPT (tibialis posterior tendon), FDLT (flexor digitorum longus tendon), 1 (medial plantar nerve), 2 (lateral plantar nerve), 3 (Baxter’s nerve), 4 (medial calcaneal branch). (**b**) shows a cranio-caudal view of the distal tarsal tunnel and its two chambers, the superior and inferior, separated by the interfascicular septum (red asterisk) and covered by the deep fascia of the ABDH (black asterisk).

**Figure 2 healthcare-12-02071-f002:**
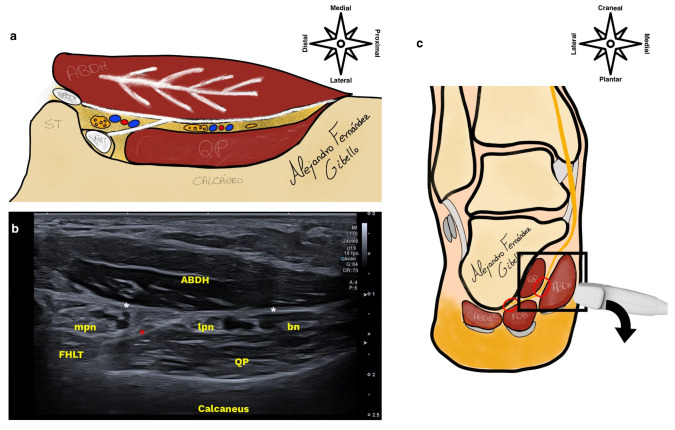
In image (**a**), you can see the illustrated version of the ultrasound in image (**b**), where we find the ABDH (abductor hallucis), the mpn (medial plantar nerve) in the upper chamber, the lpn (lateral plantar nerve) and bn (Baxter’s nerve) in the inferior chamber with the quadratus plantar (QP) in the deepest aspect, sonographically speaking, or lateral in the anatomical sense. Delimiting these structures, we find the deep fascia of the ABDH (white asterisk) and the interfascicular septum (red asterisk) both forming an “italic t”. Finally, in image (**c**), we have a coronal section of the foot and ankle where the distal tarsal tunnel is shown in a black box, and in red circles, the compression points 1 and 2 of Baxter’s nerve, of which we have only treated 1, since this is typical of the distal tarsal tunnel.

**Figure 3 healthcare-12-02071-f003:**
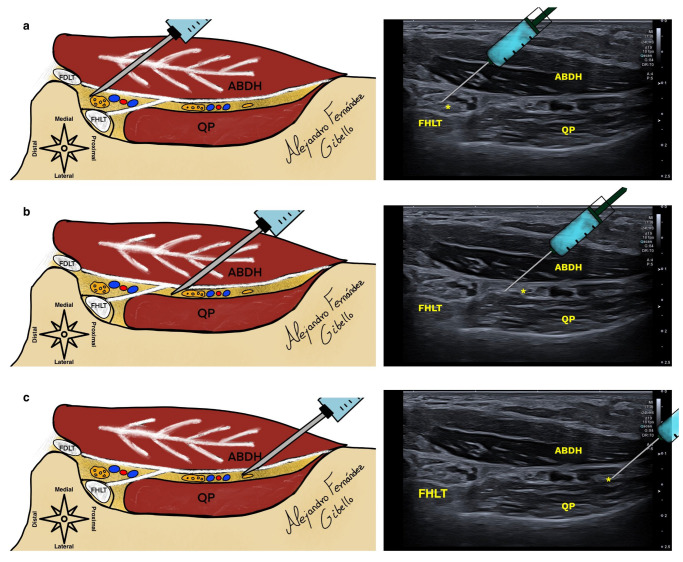
(**a**) shows the proximal–distal approach to the medial plantar nerve in the upper chamber, while (**b**) shows the lateral plantar nerve, and (**c**) illustrates Baxter’s nerve in the lower chamber. The asterisk shows the location of the medial plantar nerve, lateral plantar nerve, and inferior calcaneal nerve in images (**a**–**c**), FHLT (Flexor hallucis longus tendon).

## Data Availability

Data is contained within the article.

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
