# Peer review of "Ultrasound-Guided Approach to the Distal Tarsal Tunnel: Implications for Healthcare Research on the Medial Plantar Nerve, Lateral Plantar Nerve and Inferior Calcaneal Nerve (Baxter’s Nerve)"

_healthcare, 2024, doi:10.3390/healthcare12202071_

Round 1

Reviewer 1 Report

Comments and Suggestions for Authors Dear authors, In the paper, there are limited reference to comparative studies: It would be important to expand the literature review by comparing the effectiveness of different treatment modalities (such as radiofrequency versus other conservative treatments) in patients with TTS to provide a stronger context for the added value of the mentioned techniques.   Also, I detected some mistakes that must be corrected:
  • Line 15 ("tibial nerve"):

    • Error: The "tibial nerve" is mentioned, but later it is stated that it is often confused with the "posterior tibial nerve." This distinction could cause confusion if not clarified properly.
    • Correction: Clarify from the beginning that the correct term "tibial nerve" is used instead of "posterior tibial nerve" to avoid misunderstandings.
  • Line 39 ("superficial sensory level"):

    • Error: The phrase "superficial sensory level innervates the plantar-lateral area of the foot" may be ambiguous or poorly phrased.
    • Correction: Change to "At the superficial sensory level, the lateral plantar nerve innervates the plantar-lateral area of the foot."
  • Line 45 ("Baxter's nerve, at a sensitive level"):

    • Error: The term "sensitive" is incorrectly used to describe the sensory level.
    • Correction: Change "at a sensitive level" to "at a sensory level."
  • Line 103 ("posterior acoustic reinforcement"):

    • Error: The phrase "posterior acoustic reinforcement" may not be clear to some readers.
    • Correction: Replace with "posterior acoustic shadowing" or "posterior acoustic enhancement," depending on the intended meaning.
  • Lines 124-125 (Figure 3):

    • Error: "In image a...in image b" is not the best way to refer to the images without further description.
    • Correction: Include more detailed descriptions of the images: "Figure 3a shows the proximal-distal approach to the medial plantar nerve in the upper chamber, while Figure 3b shows the lateral plantar nerve, and Figure 3c illustrates Baxter's nerve in the lower chamber."
  • Line 144 ("Radiofrequency can be an effective treatment"):

    • Error: The phrase is too general.
    • Correction: Be more specific about the types of radiofrequency that can be used and in which cases it is most appropriate. For example, "Pulsed radiofrequency can be an effective treatment for specific cases of recalcitrant plantar fasciopathies."
  • Line 150 ("it would not be the most indicated treatment"):

    • Error: The phrase is redundant and uses incorrect terminology ("most indicated").
    • Correction: Change to "it would not be the most appropriate treatment."

Author Response

Reviewer 1’s commnents:

Dear authors, In the paper, there are limited reference to comparative studies: It would be important to expand the literature review by comparing the effectiveness of different treatment modalities (such as radiofrequency versus other conservative treatments) in patients with TTS to provide a stronger context for the added value of the mentioned techniques.   Also, I detected some mistakes that must be corrected:

    Line 15 ("tibial nerve"):

        Error: The "tibial nerve" is mentioned, but later it is stated that it is often confused with the "posterior tibial nerve." This distinction could cause confusion if not clarified properly.

        Correction: Clarify from the beginning that the correct term "tibial nerve" is used instead of "posterior tibial nerve" to avoid misunderstandings.

Response: I have changed the mentionaed test to: “The tibial nerve,  commonly misnamed as the posterior tibial nerve”

    Line 39 ("superficial sensory level"):

        Error: The phrase "superficial sensory level innervates the plantar-lateral area of the foot" may be ambiguous or poorly phrased.

        Correction: Change to "At the superficial sensory level, the lateral plantar nerve innervates the plantar-lateral area of the foot."

Response: I have changed the mentionaed test to: “on the other hand we have the lateral plantar nerve which at the superficial sensory level, the lateral plantar nerve innervates the plantar-lateral area of the foot”

    Line 45 ("Baxter's nerve, at a sensitive level"):

        Error: The term "sensitive" is incorrectly used to describe the sensory level.

        Correction: Change "at a sensitive level" to "at a sensory level."

Response: I have changed the mentionaed test to: “at a sensory level it innervates only deep structures such as the periosteum of the calcaneus”

    Line 103 ("posterior acoustic reinforcement"):

        Error: The phrase "posterior acoustic reinforcement" may not be clear to some readers.

        Correction: Replace with "posterior acoustic shadowing" or "posterior acoustic enhancement," depending on the intended meaning.

Response: I have changed the mentionaed test to: “posterior acoustic enhancement”

    Lines 124-125 (Figure 3):

        Error: "In image a...in image b" is not the best way to refer to the images without further description.

        Correction: Include more detailed descriptions of the images: "Figure 3a shows the proximal-distal approach to the medial plantar nerve in the upper chamber, while Figure 3b shows the lateral plantar nerve, and Figure 3c illustrates Baxter's nerve in the lower chamber."

Response: I have changed the mentionaed test to: “Figure 3a shows the proximal-distal approach to the medial plantar nerve in the upper chamber, while Figure 3b shows the lateral plantar nerve, and Figure 3c illustrates Baxter's nerve in the lower chamber”

    Line 144 ("Radiofrequency can be an effective treatment"):

        Error: The phrase is too general.

        Correction: Be more specific about the types of radiofrequency that can be used and in which cases it is most appropriate. For example, "Pulsed radiofrequency can be an effective treatment for specific cases of recalcitrant plantar fasciopathies."

Response: I have changed the mentionaed test to: “Pulsed radiofrequency can be an effective treatment, often used in combination with other treatments for the management of recalcitrant plantar fasciopathies, with the aim of neuromodulating or continuous radiofrequency can be an effective treatment in the ablation of the inferior calcaneal nerve for the treatment of neuropathic heel pain when Baxter's nerve compression is involved, or simply to neuromodulate if the patient has nerve involvement of the tibial nerve or one of its branches”

    Line 150 ("it would not be the most indicated treatment"):

        Error: The phrase is redundant and uses incorrect terminology ("most indicated").

        Correction: Change to "it would not be the most appropriate treatment."

Response: I have changed the mentionaed test to: “it would not be the most appropriate treatment”

Reviewer 2 Report

Comments and Suggestions for Authors The paper is very interesting, having a good clinical relevance. The connection between the described anatomy and its clinical implications in patients with heel pain of neural origin, such as tarsal tunnel syndrome (TTS), is crucial for guiding more specific treatments.   But, there are specific issues that must be discussed before the possible acceptance. The article focuses on the interventional techniques and their anatomical justification but lacks data on long-term clinical outcomes. Including follow-up studies or data on the clinical effectiveness of the proposed interventions would be essential to validate the article's proposal.   After this modification I will reassess the paper to take the proper decision

Author Response

The paper is very interesting, having a good clinical relevance. The connection between the described anatomy and its clinical implications in patients with heel pain of neural origin, such as tarsal tunnel syndrome (TTS), is crucial for guiding more specific treatments.   But, there are specific issues that must be discussed before the possible acceptance. The article focuses on the interventional techniques and their anatomical justification but lacks data on long-term clinical outcomes. Including follow-up studies or data on the clinical effectiveness of the proposed interventions would be essential to validate the article's proposal.   After this modification I will reassess the paper to take the proper decision.

Response: We have added the following

“Pulsed radiofrequency can be an effective treatment, often used in combination with other treatments for the management of recalcitrant plantar fasciopathies, with the aim of neuromodulating or continuous radiofrequency can be an effective treatment in the ablation of the inferior calcaneal nerve for the treatment of neuropathic heel pain when Baxter's nerve compression is involved, or simply to neuromodulate if the patient has nerve involvement of the tibial nerve or one of its branches. Some studies have used pulsed RF on the medial calcaneal branch to treat plantar fasciopathies, but we must remember that this is a superficial sensory branch and it does not innervate the plantar fascia, so in the absence of mononeuropathy of this branch, it would not be the most appropriate treatment (10).

Other authors perform ablative radiofrequency of the medial and inferior branches of the calcaneus to treat chronic heel pain, achieving improvement in 88% of patients (11).”

“Neural hydrodissection techniques could be an effective long-term treatment because they specifically target one of the most common causes of tarsal tunnel syndrome: nerve compression. However, it is essential to understand that the origins can be multifactorial and that the high rate of false negatives in nerve conduction studies complicates the diagnosis and prognosis of this condition. Depending on the degree of nerve injury, regeneration can be complete, partial, or virtually absent. Therefore, it would be beneficial for future research to focus on improving the neurophysiological diagnosis of this syndrome, aiming to establish a 'gold standard' test that provides a more accurate diagnosis and a better understanding of the extent of the injury.”

Reviewer 3 Report

Comments and Suggestions for Authors

Dear authors, thank you for the opportunity to review your article “Ultrasound-guided approach to the distal tarsal tunnel: implications for healthcare research on the medial plantar nerve, lateral plantar nerve and inferior calcaneal nerve (Baxter´s nerve)”. The paper presents an innovative approach to ultrasound-guided anatomy: The use of ultrasound to precisely visualize the nerve structures within the distal tarsal tunnel is a key strength of the study. The focus on the accurate identification of the involved nerves provides a breakthrough in improving interventional treatments.

However, I have detected some areas of improvement. The paper could improve if you address the following issues:

The article mentions the diagnostic challenges of TTS due to false negatives in neurophysiological studies but does not elaborate on practical solutions to address these limitations. A more detailed discussion of alternative diagnostic methods that could complement ultrasound and improve diagnostic accuracy, such as advanced imaging studies or a combination of other clinical techniques, would be beneficial. Further discussion on diagnostic limitations is needed.

Although the difficulty of classifying the degree of nerve injury in the tarsal tunnel is mentioned, the article does not explore how future research might address this issue. A more extensive discussion of how neurophysiological studies and new approaches, such as neurostimulation, could help standardize these classifications would be valuable.

Hope that you find these suggestions as a chance to improve the whole quality of the paper.

Author Response

Reviewer 3’s commnents:

Dear authors, thank you for the opportunity to review your article “Ultrasound-guided approach to the distal tarsal tunnel: implications for healthcare research on the medial plantar nerve, lateral plantar nerve and inferior calcaneal nerve (Baxter´s nerve)”. The paper presents an innovative approach to ultrasound-guided anatomy: The use of ultrasound to precisely visualize the nerve structures within the distal tarsal tunnel is a key strength of the study. The focus on the accurate identification of the involved nerves provides a breakthrough in improving interventional treatments.

However, I have detected some areas of improvement. The paper could improve if you address the following issues:

The article mentions the diagnostic challenges of TTS due to false negatives in neurophysiological studies but does not elaborate on practical solutions to address these limitations. A more detailed discussion of alternative diagnostic methods that could complement ultrasound and improve diagnostic accuracy, such as advanced imaging studies or a combination of other clinical techniques, would be beneficial. Further discussion on diagnostic limitations is needed.

Response: We have added de the following

“It is an under-diagnosed condition because neurophysiological studies have an estimated false negative rate of 50% (4).According to the American Association of Neuromuscular and Electrodiagnostic Medicine, you could diagnose TTS if the patient has neuropathic signs or symptoms with a positive Tinel's sign (5).

However, there are other methods that can assist in the diagnosis of this condition: ultrasound allows us to assess the echostructure of the nerve, and we can also evaluate the cross-sectional area of the nerve before entering the tunnel and within the tunnel as a diagnostic sign”

Although the difficulty of classifying the degree of nerve injury in the tarsal tunnel is mentioned, the article does not explore how future research might address this issue. A more extensive discussion of how neurophysiological studies and new approaches, such as neurostimulation, could help standardize these classifications would be valuable.

Response:

We are currently working on it, and we would like to have it ready as soon as possible to be able to show it to you and send you a new article for review that improves the sensitivity of neurophysiological studies in the tarsal tunnel. This is why we mention in the study that, among other issues, in neurophysiological studies, only 2 of the 4 branches are examined in most cases, which represents 50% of them. Your observation is excellent, as from our point of view, this pathology is so underdiagnosed due to the high rate of false negatives.

“Tarsal tunnel syndrome is a complicated pathology to diagnose because, as we have already mentioned, it has a large number of false negatives and the treatments have a low rate of effectiveness because if we do not know the number of branches affected, the degree of injury to them, etc., then we cannot give the patient the best treatment. This is why more research is needed into neurophysiological protocols that allow us to obtain a more accurate diagnosis, as some authors do (9), but we must not forget that if we only study the medial and lateral plantar branches, we are only studying 50% of the branches and we are ignoring the inferior calcaneal nerve and the medial calcaneal branch”

Hope that you find these suggestions as a chance to improve the whole quality of the paper.